# High Oxygen Does Not Increase Reperfusion Injury Assessed with Lipid Peroxidation Biomarkers after Cardiac Arrest: A Post Hoc Analysis of the COMACARE Trial

**DOI:** 10.3390/jcm10184226

**Published:** 2021-09-17

**Authors:** Jaana Humaloja, Maximo Vento, Julia Kuligowski, Sture Andersson, José David Piñeiro-Ramos, Ángel Sánchez-Illana, Erik Litonius, Pekka Jakkula, Johanna Hästbacka, Stepani Bendel, Marjaana Tiainen, Matti Reinikainen, Markus B. Skrifvars

**Affiliations:** 1Department of Emergency Care and Services, University of Helsinki and Helsinki University Hospital, 00029 Helsinki, Finland; jaana.humaloja@outlook.com; 2Neonatal Research Group, Health Research Institute La Fe, 46026 Valencia, Spain; maximo.vento@uv.es (M.V.); julia.kuligowski@uv.es (J.K.); jose_pineiro@iislafe.es (J.D.P.-R.); asanchezillana@gmail.com (Á.S.-I.); 3Pediatric Research Center, Children’s Hospital, University of Helsinki and Helsinki University Hospital, 00029 Helsinki, Finland; sture.andersson@hus.fi; 4Department of Anesthesiology, Intensive Care and Pain Medicine, University of Helsinki and Helsinki University Hospital, 00029 Helsinki, Finland; erik.litonius@hus.fi (E.L.); pekka.jakkula@me.com (P.J.); johanna.hastbacka@hus.fi (J.H.); 5Department of Anesthesiology and Intensive Care, University of Eastern Finland and Kuopio University Hospital, 70029 Kuopio, Finland; stepani.bendel@kuh.fi (S.B.); matti.reinikainen@kuh.fi (M.R.); 6Department of Neurology, University of Helsinki and Helsinki University Hospital, 00029 Helsinki, Finland; marjaana.tiainen@hus.fi

**Keywords:** out-of-hospital cardiac arrest, isoprostanes, neuroprostanes, lipid peroxides, reperfusion injury

## Abstract

The products of polyunsaturated fatty acid peroxidation are considered reliable biomarkers of oxidative injury in vivo. We investigated ischemia-reperfusion-related oxidative injury by determining the levels of lipid peroxidation biomarkers (isoprostane, isofuran, neuroprostane, and neurofuran) after cardiac arrest and tested the associations between the biomarkers and different arterial oxygen tensions (PaO_2_). We utilized blood samples collected during the COMACARE trial (NCT02698917). In the trial, 123 patients resuscitated from out-of-hospital cardiac arrest were treated with a 10–15 kPa or 20–25 kPa PaO_2_ target during the initial 36 h in the intensive care unit. We measured the biomarker levels at admission, and 24, 48, and 72 h thereafter. We compared biomarker levels in the intervention groups and in groups that differed in oxygen exposure prior to randomization. Blood samples for biomarker determination were available for 112 patients. All four biomarker levels peaked at 24 h; the increase appeared greater in younger patients and in patients without bystander-initiated life support. No association between the lipid peroxidation biomarkers and oxygen exposure either before or after randomization was found. Increases in the biomarker levels during the first 24 h in intensive care suggest continuing oxidative stress, but the clinical relevance of this remains unresolved.

## 1. Introduction

During return of spontaneous circulation after cardiac arrest and global ischemia, some degree of reperfusion injury is inevitable [1]. One mechanism of oxidative stress is the formation of reactive oxygen species (ROS). Experimental evidence suggests that higher PaO_2_ in blood and tissues enhances ROS formation [2]. ROS are major mediators of damage to tissue components, such as proteins, lipids, and DNA, in ischemia-reperfusion. Moreover, ROS trigger the activation of secondary injury pathways, including inflammation, autophagy, and apoptosis [3]. Cell membrane structures highly concentrated with fatty acids are sensitive to oxidative injury, and the metabolic byproducts of the injury can be measured in the blood. Measurable oxidative injury lipid biomarkers produced in vivo include isoprostanes (IsoPs) and isofurans (IsoFs) derived from arachidonic acid, neuroprostanes (NeuroPs) and neurofurans (NeuroFs) derived from docosahexaenoic acid, and di-homo-isoprostanes (di-homo-IsoPs) and di-homo-isofurans (di-homo-IsoFs) derived from adrenic acid [4,5,6]. These biomarkers are considered relevant surrogates of oxidative stress-derived injury, and increased levels of these biomarkers can be detected in many pathological conditions [7,8]. Arachidonic acid can be found in any tissue; accordingly, the levels of IsoPs and IsoFs reflect the total oxidative stress the body is being subjected to. Docosahexaenoic acid, conversely, is highly abundant in central nervous tissue grey matter, and adrenic acid is a main component of brain white matter. Accordingly, NeuroPs, NeuroFs, di-homo-IsoPs, and di-homo-IsoFs have been found by experiments to be specific indicators of oxidative injury in nervous tissue [8].

These lipid peroxidation biomarkers have been successfully used to measure oxidative injury arising immediately after resuscitation in neonates; whether they could be relevant after adult cardiac arrest is unknown [9]. In this post hoc study of the COMACARE trial (NCT02698917), we measured the plasma levels of biomarkers related to lipid peroxidation (IsoPs, IsoFs, NeuroPs, NeuroFs, di-homo-IsoPs, and di-homo-IsoFs) in adult patients after out-of-hospital cardiac arrest. The trial randomized patients to two different PaO_2_ targets during the first 36 h of intensive care treatment [10]. Our aim was to test the associations between lipid peroxidation biomarker levels and PaO_2_ and the fractions of inspired oxygen (FiO_2_) used. The associations between biomarkers and oxygenation were tested in three settings: prehospital, on intensive care unit (ICU) admission, and during the first 24 h of ICU treatment. Based on experimental evidence, we hypothesized that higher blood oxygen and higher FiO_2_ would be associated with higher lipid peroxidation biomarker levels [11,12,13].

## 2. Materials and Methods

### 2.1. Trial Design, Participants, and Inclusion Criteria

This study was based on post hoc analysis of arterial blood samples collected during the COMACARE trial (NCT02698917), March 2016–November 2017. The trial protocol and main findings were published previously in detail [14]. The Northern Savo Hospital District research ethics committee approved the study plan (decision number 295/13.02.00/2015, 23 February 2016), and this study was conducted in accordance with the Declaration of Helsinki. In brief, we randomized 123 patients resuscitated from out-of-hospital cardiac arrest with 2^3^ factorial design: normal or moderately elevated arterial oxygen tension (PaO_2_ 10–15 kPa or 20–25 kPa), low-normal or high-normal arterial carbon dioxide tension (PaCO_2_ 4.5–4.7 kPa or 5.8–6.0 kPa), and low-normal or high-normal mean arterial pressure (65–75 mmHg or 80–100 mmHg) for the first 36 h of ICU treatment. All patients were treated according to the current guidelines and underwent targeted temperature therapy (TTM) at 33 or 36 °C. The participants included adult (aged 18–80) patients resuscitated from witnessed out-of-hospital cardiac arrest, with initially shockable rhythm, and with markedly impaired consciousness, and who were mechanically ventilated on ICU admission. The patients in this sub-study of the COMACARE trial were from six different ICUs in Finland. The flowchart of patient inclusion is in Appendix A.

### 2.2. Blood Samples and Sample Analysis

Blood samples for biomarker determination were collected during the trial and frozen immediately after centrifugation. We determined the relative levels of isoprostanes, isofurans, neuroprostanes, neurofurans, di-homo-isoprostanes, and di-homo-isofurans from samples collected on ICU admission (0 h) and 24, 48, and 72 h after admission. The samples were analyzed in May 2020 in the Analytical Unit, Health Research Institute La Fe, Valencia, Spain, using ultra-performance liquid chromatography tandem mass spectrometry [15,16]. A specific description of the analysis method can be found in the Appendix A.

### 2.3. Outcome Measures

The primary outcome measures were lipid peroxidation biomarker levels (IsoPs, IsoFs, NeuroPs, NeuroFs, di-homo-IsoPs, and di-homo-IsoFs) at 0 and 24 h. Lipid peroxides remain stable in the blood, but with normal glomerular filtration are rapidly secreted in urine; therefore, we were mainly interested biomarkers measured at earlier time-points [17]. The secondary outcome was the patient’s functional outcome at 6 months after the arrest determined with the Cerebral Performance Category [18]. A favorable functional outcome was defined as a Cerebral Performance Category (CPC) score of 1 or 2, corresponding to independence in daily activities as a minimum; and an unfavorable outcome as a CPC score of 3–5, corresponding to a spectrum from severe cerebral disability to death.

### 2.4. Exposure to Oxygen

We determined oxygen exposure in three settings: prehospital, on ICU admission, and during the first 24 h of ICU care. We obtained prehospital PaO_2_ from the first arterial sample collected after return of spontaneous circulation (ROSC) obtained by a physician-staffed ambulance using a point-of-care device. On ICU admission, we determined oxygenation from the first PaO_2_ measured and the simultaneously used FiO_2_ after arrival at the ICU. Finally, we determined the highest PaO_2_, the PaO_2_ area under the curve, and the FiO_2_ area under the curve over the first 24 h of ICU treatment. All arterial blood gases determined in the ICU were analyzed at the site with point-of-care device. The timeline of the study protocol is in the Appendix A.

### 2.5. Statistical Methods

We present continuous data as medians and interquartile ranges (IQR) and categorical data as counts and percentages. We tested all continuous variables for normality and used the Mann–Whitney U test to compare the non-normally distributed data [19]. We compared categorical data with Pearson’s Chi-squared test [20]. The associations between oxygen exposure and biomarker levels were analyzed in scatterplot visualizations and with linear regression and linear mixed model analysis [21,22]. We performed Log transformations for the biomarker concentration values. The method for building a linear regression model and more detailed descriptions of the statistical methods used can be found in the Appendix A. Statistical analyses were performed with IBM SPSS 27.0.1.0 and GraphPad Prism 9.0.1. for MacOS.

## 3. Results

The study included 112 patients with blood samples available for analysis. A flowchart of patient inclusion in every sub-analysis is in the Appendix A. The di-homo-IsoPs’ and di-homo-IsoFs’ concentrations were below the limit of detection and were therefore not included in the data analysis.

We divided patients based on the median value of the first ICU-measured PaO_2_ (≤15.60 vs. > 15.60 kPa) and the corresponding used FiO_2_ (<0.50 vs. ≥0.50). The patients with lower admission PaO_2_ had a higher median body mass index (IQR) of 27.8 (23.7–30.9) vs. 25.5 (23.8–27.8). Longer delay to arrival of the first unit (8 [6,7,8,9,10] vs. 6.5 [5,6,7,8,9] minutes) was associated with lower admission PaO_2_. The baseline characteristics of the study population, a comparison between low and high admission PaO_2_, and comparison between the randomized PaO_2_ groups are shown in Table 1.

### 3.1. Biomarker Levels on Admission (0 h) and Prehospital Oxygen Exposure

Prehospital arterial samples were available for 54 patients. There were no associations between the prehospital PaO_2_ and the biomarker levels determined on ICU admission (Appendix A). The prehospital PaO_2_ and PaO_2_ values on ICU admission were not significantly correlated; the Pearson correlation coefficient was 0.24 (*p* = 0.08). In the higher admission PaO_2_ (>15.6 kPa) group, the median IsoFs levels at 0 h were slightly higher compared to the lower admission PaO_2_ group (*p* = 0.04). There were no clear correlations between the admission PaO_2_ and the other biomarkers (Figure 1). In the higher admission FiO_2_ (≥0.50) group the median IsoFs level at 0 h was slightly lower compared to the lower admission FiO_2_ group (*p* = 0.04). There was no difference between the admission FiO_2_ and the other biomarker levels on admission (Figure 2).

The time from the arrest to ICU admission varied from 58 to 385 min; median (IQR) 147 (104–195). There were no associations between the time from the arrest to ICU admission and biomarker levels on admission (Appendix A). Additionally, there were no differences in biomarker levels on admission between those arriving to the ICU early (admission < 147 min after the arrest) compared to the later admissions (Appendix A).

### 3.2. Biomarker Levels at 24 h and Oxygen Exposure

The biomarker levels at 24 h were not significantly different among the randomized oxygen target groups. There was no clear difference between any biomarker level at 24 h by the highest PaO_2_, the PaO_2_ area under the curve, or the FiO_2_ area under the curve (Appendix A).

### 3.3. Biomarker Levels over Time

The time profile for every biomarker level at 0, 24, 48, and 72 h can be seen in Figure 3. All 112 patients underwent analytical determinations; however, for some patients, biomarker values are missing at single time points. The time profiles were similar for all measured biomarkers. The highest levels were seen at 24 h. In the linear mixed model, there were no significant differences in any of the biomarker levels over time between the randomized PaO_2_ groups (Time*PaO_2_—column in the Appendix A; the *p*-values are also presented in Figure 3). When comparing the biomarker levels at all time points individually between the patients with favorable outcomes and those with unfavorable outcomes, the IsoFs, IsoPs, and NeuroFs values were found to be higher on arrival in the patients with subsequent favorable functional outcomes, but the difference was significant only regarding IsoPs (Appendix A). The median (IQR) IsoPs level in the favorable outcome groups was 3.30 (2.06–5.04) compared to 2.12 (1.18–3.23) procedure defined units ([p.d.u.]) in the unfavorable outcome group; *p* = 0.006. A linear mixed model analysis showed no significant differences in biomarker levels over time for the IsoPs, IsoFs, and NeuroFs, but the NeuroPs levels were significantly different over time between the favorable and unfavorable functional outcome groups (Appendix A).

### 3.4. Linear Regression Analysis

In the linear regression models to predict biomarker levels at 0 and 24 h, 103 and 109 patients, respectively, were included in the analyses. Higher age indicated lower biomarker levels in all four biomarkers at 24 h and lower levels of IsoPs and IsoFs at 0 h. Bystander resuscitation was associated with lower IsoPs, NeuroPs, and NeuroFs levels at 24 h. Higher FiO_2_ on arrival to the ICU was associated with higher NeuroFs levels at 0 h. The results for the linear regression analyses for biomarker levels at 24 h are shown in Table 2; the biomarker level results at 0 h are presented in the Appendix A. The predictive power of the regression models remained low; the R-squared for every linear regression model was below 0.15.

## 4. Discussion

We explored the levels of lipid peroxidation biomarkers previously conjoined with oxidative ischemia-reperfusion injury and their associations with oxygen exposure in patients after out-of-hospital cardiac arrest. Contrary to our hypothesis, we did not find any significant associations between exposure to different partial pressures of arterial oxygen or FiO_2_ and the measured biomarkers. These biomarkers have experimentally and in newborns been shown to reflect reperfusion injury to the brain and other organs; however, in our study, we did not find any connection between oxygenation and the biomarkers either early after cardiac arrest or during ICU care [9,12,23,24].

Observational studies have suggested increased mortality in patients subjected to an excess of oxygen upon reperfusion and reported the generation of oxygen free radicals as a potential mechanism for worsening reperfusion injury [25,26]. A recent meta-analysis found evidence supporting conservative oxygen therapy after CA [27]. Preterm neonates have immature antioxidant capacity, which makes them vulnerable to oxidative stress, and an association between oxygen use and lipid peroxidation has been shown [9,28]. Lipid peroxidation biomarkers have been the key determinants in the development of neonatal resuscitation policies [29]. Several clinical trials have shown increased mortality, greater oxidative stress, and higher probability for hypoxic-ischemic encephalopathy when using FiO_2_ 1.00 vs. FiO_2_ 0.21 during newborn resuscitation [23,29].

We failed to demonstrate an increase in lipid peroxidation with higher vs. lower oxygen exposure in our adult cardiac arrest patients; the noticed difference in the median IsoFs levels between higher and lower PaO_2_ on ICU admission was minor and likely clinically irrelevant. Our data allow us only to speculate about possible explanations to our findings. ROS-mediated injury is the consequence of the exacerbated chemical reactivity of free radicals, and therefore, it occurs immediately upon reperfusion. In this scenario, it is plausible that during the time elapsed between emergency stabilization and arrival at the ICU, the bulk of oxidative byproducts generated were washed out from the circulation. Consequently, a greater difference in PaO_2_ levels during intensive care would have been required to elicit differences in lipid peroxidation [30,31]. Hypothermia reduces the mitochondrial respiratory rate and decreases ROS generation [32]. All patients underwent targeted temperature treatment, which might attenuate the oxidative stress and the ensuing lipid peroxidation [32,33,34,35]. The median time from arrest to ICU admission was over two hours, due to resuscitation efforts and the need for coronary angiography; hence, cellular adaptation to manage oxidative stress might have occurred by the time the patients reached the ICU. Additionally, the extent of the oxidative injury occurring immediately after the return of spontaneous circulation could mask all the differences thereafter. The early increase in biomarker levels during the first 24 h in the ICU probably reflects a second oxidative stress burst enhanced by the oxygen supplementation that ensued on arrival at the ICU. The subsequent increase in the biomarker levels on arrival at the ICU could arise from poorly perfused tissues coinciding with hemodynamic stabilization in the ICU, e.g., liver and kidneys, or it could reflect depletion of the body’s antioxidant capacity due to the high amount of ROS generated during the initial reperfusion.

During cardiac arrest, the duration of ischemia is the major determinant of ischemic cell injury and contributes to the accumulation of purine derivatives [3,30]. Upon reperfusion, xanthine oxidoreductase and specific proteases in the presence of oxygen metabolize purine derivatives to anion superoxide and hydroxyl radicals that cause structural and functional damage to cell components [13,36,37,38]. The weak association observed between lower biomarker levels at 24 h and bystander cardiopulmonary resuscitation suggests that a longer duration of no-flow period might increase vulnerability to oxidative stress after reperfusion. Several animal studies have detected more extensive ROS-mediated injuries and increased lipid peroxidation when higher FiO_2_ was used following resuscitation [12,24,39]. Liu et al. showed that hyperoxic ventilation (FiO_2_ 1.00 vs. 0.21) during the early post-resuscitation period in dogs leads to increased amounts of oxidated lipids in the frontal cortex and a poorer functional outcome at 24 h [12]. One rat study detected increased myocardium IsoPs levels, worse myocardial function, and shorter duration of survival when rats were ventilated with FiO_2_ 1.00 compared to lower FiO_2_ levels during resuscitation and in the early post-resuscitation period [24]. Solberg et al. showed in a piglet model that the amount of oxygen used during resuscitation correlated dose-dependently with the urinary markers of oxidative damage to DNA and amino acids [39]. IsoPs and IsoFs are generated after oxidative damage in all tissues, but NeuroFs and NeuroPs are mostly generated in the brain’s grey matter [4,6]. The brain has a high concentration of polyunsaturated fatty acids compared to other organs, and thus lipid peroxidation is the main outcome of ROS-induced brain tissue injury [1]. In adults, higher levels of NeuroPs have been measured in cerebrospinal fluid after traumatic brain injury and subarachnoid hemorrhage [40]. Generation of NeuroFs is dependent on brain tissue oxidation; NeuroPs are generated due to oxidative stress irrespective of brain oxygen level. Based on our study, we cannot conclude that these biomarkers’ levels correlate with the extent of the actual reperfusion injury. Better functional outcomes (cerebral performance categories 1–2) were associated with slightly higher levels of lipid peroxidation biomarkers; one reason could be that younger patients, who have a tendency to recover with better functional outcomes, exhibit more pronounced responses to oxidative stress, increasing lipid peroxidation. Unfortunately, we were not able to detect di-homo-IsoPs or di-homo-IsoFs, which are released in white matter injury [15,41].

Our study has several strengths. To assess oxidative injury during the early phases of post-resuscitation care in the ICU, we included two groups with distinct oxygen treatment targets, and the treatment targets were well maintained during the intervention period [10]. There is evidence that the studied biomarkers reliably reflect ROS-mediated cell injury in vivo, the method used for biomarker analysis is very sensitive, and lipid peroxides stay stable in the samples when stored correctly [4,5,6,9,42,43,44]. Our study was the first to measure values in the setting of adult cardiac arrest. Our study has some limitations. The timing of the collected blood samples was not planned based on the features of the studied biomarkers. We lacked samples taken prior to ICU admission, which would reflect oxidative stress immediately after reperfusion [45,46]. Additionally, more frequent sampling would improve understanding how these markers react during the early hours of intensive care. Only half of the study population had the prehospital PaO_2_ data available. We note that our sample size may have been too small to detect a difference in lipid peroxidation between the oxygen exposure groups. However, studies conducted in animals and in neonates have shown effect sizes of a magnitude that we would have been able to detect if they were present [11,24,47]. Finally, due to the study setting, we could not assess lipid peroxidation in specific tissues (e.g., the brain).

## 5. Conclusions

Lipid peroxidation biomarkers were similar, irrespective of arterial oxygen levels early after cardiac arrest, and no differences in biomarker levels were found after treatment with two different arterial oxygen targets during ICU care. The increases in lipid peroxidation biomarkers during the first 24 h may reflect a second burst of oxidative stress in the ICU, but the clinical relevance of these increases remains unresolved.

## Figures and Tables

**Figure 1 jcm-10-04226-f001:**
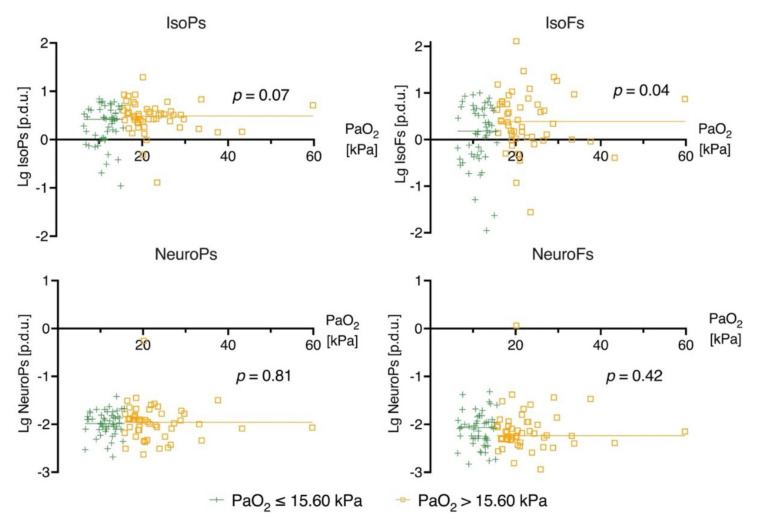
Scatterplot of biomarker and PaO_2_ levels on ICU admission (0 h). The patients are divided into two groups by median PaO_2_ value on intensive care unit admission. The straight lines seen in the plots represent the median values of the biomarker levels concerned. The *p*-value indicates the significance of the Mann–Whitney U test between the biomarker level on admission in the patients’ PaO_2_ below vs. above the median (15.60 kPa). Definitions of abbreviations: [p.d.u.] = procedure defined unit; Lg = base 10 logarithm.

**Figure 2 jcm-10-04226-f002:**
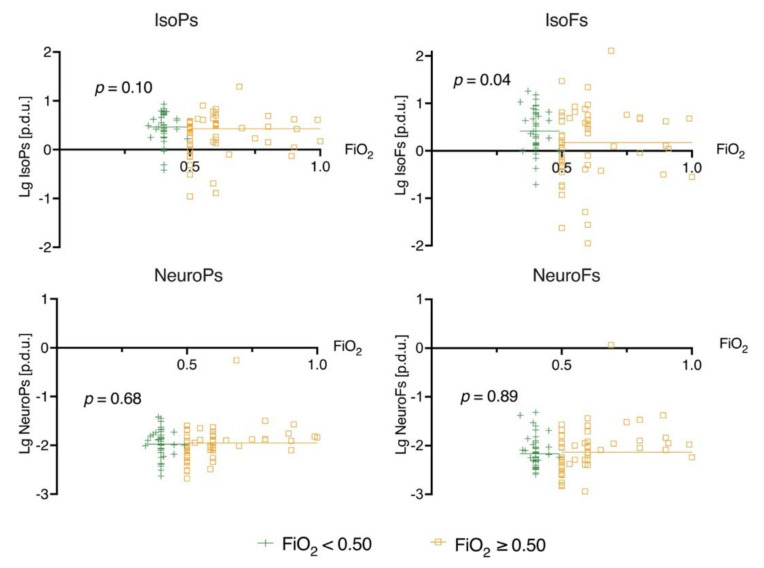
Scatterplot of biomarker and FiO_2_ levels on ICU admission (0 h). The patients are divided into two groups by the median FiO_2_ value on intensive care unit admission. The straight lines seen in the plots represent the median values of the biomarkers concerned. The *p*-value indicates the significance of the Mann–Whitney U test between the biomarker level on admission in the patients’ FiO_2_ below vs. above the median (0.50). Definitions of abbreviations: [p.d.u.] = procedure defined unit; Lg = base 10 logarithm.

**Figure 3 jcm-10-04226-f003:**
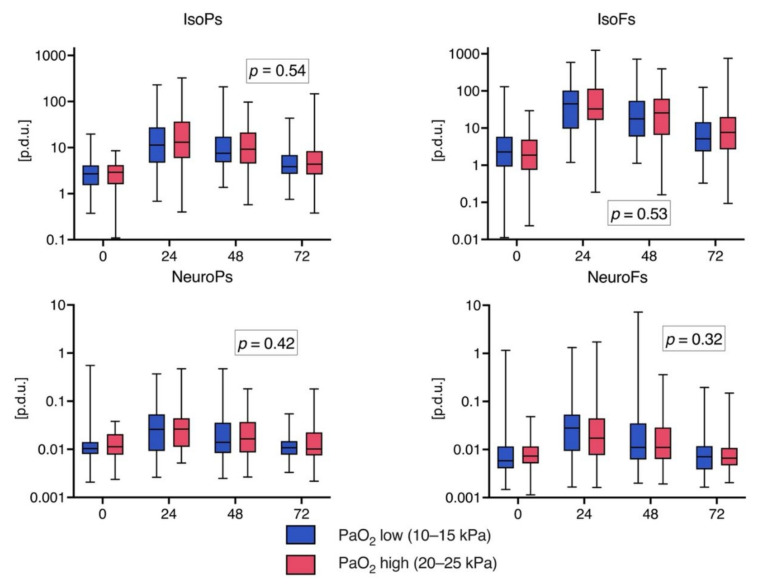
Median biomarker levels on ICU admission and 24, 48, and 72 h after admission. The midline of the box represents the median; the upper and lower borders of the box represent the interquartile range. The whiskers are at the minimum and maximum values. Note the logarithmic scale on the y-axis. The *p*-value indicates the significance of the biomarker level change over time between the low vs. high PaO_2_ treatment target groups determined with linear mixed model analysis. Definitions of abbreviations: [p.d.u] = procedure defined unit.

**Table 1 jcm-10-04226-t001:** Baseline characteristics of complete study population, study population divided by lower vs. higher admission PaO_2_, and study population in randomized oxygen groups.

		PaO_2_ on Intensive Care Unit Admission			Randomized PaO_2_ Target Groups		
	All *n* = 112	≤15.6 kPa, *n* = 56	>15.6 kPa, *n* = 56	*p*-Value ^1^	10–15 kPa, *n* = 57	20–25 kPa, *n* = 55	*p*-Value ^1^
	Median (IQR)/count (%)				Median (IQR)/count (%)		
Age (in years)	62 (53–68)	65 (54–69)	58 (52–67)	0.12	62 (53–68)	62 (54–69)	0.69
Male gender	92 (82)	43 (77)	49 (88)	0.14	47 (83)	45 (82)	0.93
BMI	26.3 (23.8–29.4)	27.8 (23.7–30.9)	25.5 (23.8–27.8)	0.015	26.2 (23.4–29.1)	26.5 (24.2–30.9)	0.39
Smoker (yes) ^2^	35 (35%)	15 (29)	20 (42)	0.20	17 (34)	18 (37)	0.78
Bystander resuscitation (yes)	93 (83)	48 (86)	45 (80)	0.45	47 (83)	46 (84)	0.87
ROSC time (min)	21 (16–26)	22 (17–28)	21 (15–25)	0.12	20 (16–25)	22 (17–27)	0.38
Delay to first unit (min)	7 (6–9)	8 (6–10)	6.5 (5–9)	0.008	7 (6–9)	7 (6–9)	0.68
30-day mortality	36 (32)	19 (34)	17 (30)	0.69	17 (30)	19 (35)	0.59
180-day mortality	37 (33)	20 (36)	17 (30)	0.55	17 (30)	20 (36)	0.46
CPC 1–2 at 180 days	73 (65)	35 (63)	38 (68)	0.55	39 (69)	34 (62)	0.46

Definitions of abbreviations: BMI = body mass index; CPC = cerebral performance category; IQR = interquartile range; ROSC = return of spontaneous circulation. ^1^ *p*-values are conducted with the Mann–Whitney U test for non-normal continuous data and with Pearson’s Chi-squared test with categorical data. ^2^ Thirteen patients were missing smoking status data.

**Table 2 jcm-10-04226-t002:** Linear regression analysis to predict biomarker levels at 24 h.

	*R* ^2^	B	95% CI for B		*p*-Value
			Lower Bound	Upper Bound	
**Lg IsoFs at 24 h**	0.08				
Age (years)		−0.01	−0.02	0	0.045
Bystander CPR		−0.22	−0.48	0.05	0.10
PaO_2_ (10–15 kPa/20–25 kPa)		0.12	−0.07	0.31	0.22
PaCO_2_ (4.5–4.7 kPa/5.8–6.0 kPa)		−0.11	−0.30	0.08	0.25
MAP (65–75 mmHg/80–100 mmHg)		0.01	−0.18	0.21	0.89
**Lg IsoPs at 24 h**	0.12				
Age (years)		−0.01	−0.02	−0.004	0.006
Bystander CPR		−0.37	−0.70	−0.04	0.03
PaO_2_		0.05	−0.19	0.29	0.68
PaCO_2_		−0.12	−0.36	0.12	0.34
MAP		0.06	−0.18	0.31	0.61
**Lg NeuroPs at 24 h**	0.12				
Age (years)		−0.01	−0.02	0	0.04
Bystander CPR		−0.36	−0.59	−0.11	0.005
PaO_2_		0.01	−0.17	0.18	0.96
PaCO_2_		0.02	−0.16	0.20	0.82
MAP		0.09	−0.09	0.27	0.34
**Lg NeuroFs at 24 h**	0.14				
Age (years)		−0.01	−0.02	−0.001	0.04
Bystander CPR		−0.53	−0.84	−0.22	0.001
PaO_2_		−0.14	−0.37	0.08	0.21
PaCO_2_		−0.07	−0.30	0.15	0.53
MAP		0.04	−0.20	0.27	0.76

Definitions of abbreviations: CPR = cardiopulmonary resuscitation; Lg = base 10 logarithm; isoFs = isofurans; isoPs = isoprostanes; MAP = mean arterial pressure; neuroFs = neurofurans; neuroPs = neuroprostanes; PaCO_2_ = partial pressure of arterial carbon dioxide; PaO_2_ = partial pressure of oxygen.

## Data Availability

The data presented in this study are available only on reasonable request from the corresponding author. The data are not publicly available due the small sample size and patient privacy concerns. The study group followed the EU General Data Protection Regulation principles.

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
