# Peer review of "High Oxygen Does Not Increase Reperfusion Injury Assessed with Lipid Peroxidation Biomarkers after Cardiac Arrest: A Post Hoc Analysis of the COMACARE Trial"

_jcm, 2021, doi:10.3390/jcm10184226_

Round 1
Reviewer 1 Report
This is a post-hoc study of the COMACARE trial examining plasma levels of biomarkers related to lipid peroxidation which would reflect reperfusion injury after ROSC and following randomization to two different oxygen tension targets during the first 24h in the ICU.
Despite sound evidence from animal model and neonates no association between the lipid peroxidation biomarkers and oxygen exposure either before or after randomization was found.
The Authors should be complimented for the effort in adding more insight to a hot topic in post resuscitation care. Randomized data is scant in this specific context.
The study is well designed and reads well.
The Authors report unexpected findings. This is often more intriguing and might lead to increase the overall knowledge on the topic than what expected and hypothesized when designing the trial. I thus encourage the Authors on continuing investigating the topic based on the present unexpected findings.
I don’t have major concerns.
Speculating on the reasons that might explain the absence of difference in the two groups of PaO2 levels the Authors suggest that perhaps a greater difference in PaO2 targets might have been necessary to elicit appreciable differences in lipid peroxidation. I suggest to consider if outliner patients with extreme PaO2 values might provide insight as proof of concept. If such data is available might be worth considering to report it. I leave this call up to the Authors
Reviewer 2 Report
Humaloj et al. tested the dependence of high oxygen following cardiac arrest on reperfusion injury by testing the levels of biomarkers at different time points following hospital admission. The authors did a splendid work and presented all the data and methodology in a very scientific way. I recommend this paper to be published but I have a few questions below:
(1) Why did the authors did not choose any time point between 0 hours and 24 hours. Maybe a timepoint at 12 hours would have given a better understanding. Maybe there was a difference in the level of biomarkers at around 12 hours which vanished by 24 hours time point?
(2) Is it possible that people of different weight (or BMI) respond differently to different pressures of oxygen?
(3) Can the authors provide some literature to justify the use of their oxygen pressures? For example, studies which show oxidative stress injury among people administered with similar pressures of oxygen.
Author Response
Please see the attachment below
